# Isolation and Characterization of a Novel *Klebsiella pneumoniae* N4-like Bacteriophage KP8

**DOI:** 10.3390/v11121115

**Published:** 2019-12-02

**Authors:** Vera Morozova, Igor Babkin, Yuliya Kozlova, Ivan Baykov, Olga Bokovaya, Artem Tikunov, Tatyana Ushakova, Alevtina Bardasheva, Elena Ryabchikova, Ekaterina Zelentsova, Nina Tikunova

**Affiliations:** 1Laboratory of Molecular Microbiology, Institute of Chemical Biology and Fundamental Medicine SB RAS, Novosibirsk 630090, Russia; i_babkin@mail.ru (I.B.); ulona@ngs.ru (Y.K.); ivan_baykov@mail.ru (I.B.); olga.vasilievna@inbox.ru (O.B.); arttik@mail.ru (A.T.); ushakova@niboch.nsc.ru (T.U.); herba12@mail.ru (A.B.); lenryab@niboch.nsc.ru (E.R.); zelentsova@tomo.nsc.ru (E.Z.); tikunova@niboch.nsc.ru (N.T.); 2Novosibirsk State University, Novosibirsk 630090, Russia; 3International Tomography Center Siberian Branch of Russian Academy of Sciences (SB RAS), Novosibirsk 630090, Russia

**Keywords:** bacteriophage, *Klebsiella pneumoniae*, N4-like podovirus, *Enquatrovirus*, *Gamaleyavirus*

## Abstract

*Klebsiella pneumoniae* is a common pathogen, associated with a wide spectrum of infections, and clinical isolates of *K. pneumoniae* often possess multiple antibiotic resistances. Here, we describe a novel lytic N4-like bacteriophage KP8, specific to *K. pneumoniae*, including its genome, partial structural proteome, biological properties, and proposed taxonomy. Electron microscopy revealed that KP8 belongs to the Podoviridae family. The size of the KP8 genome was 73,679 bp, and it comprised 97 putative open reading frames. Comparative genome analysis revealed that the KP8 genome possessed the highest similarity to the genomes of *Enquatrovirus* and *Gamaleyavirus* phages, which are N4-like podoviruses. In addition, the KP8 genome showed gene synteny typical of the N4-like podoviruses and contained the gene encoding a large virion-encapsulated RNA polymerase. Phylogenetic analysis of the KP8 genome revealed that the KP8 genome formed a distinct branch within the clade, which included the members of *Enquatrovirus* and *Gamaleyavirus* genera besides KP8. The average evolutionary divergences KP8/*Enquatrovirus* and KP8/*Gamaleyavirus* were 0.466 and 0.447 substitutions per site (substitutes/site), respectively, similar to that between *Enquatrovirus* and *Gamaleyavirus* genera (0.468 substitutes/site). The obtained data suggested that *Klebsiella* phage KP8 differs from other similar phages and may represent a new genus within the N4-like phages.

## 1. Introduction

*Klebsiella pneumoniae* is a Gram-negative, non-motile, non-sporulating bacterium belonging to the *Klebsiella* genus of the Enterobacteriaceae family. Members of the *Klebsiella* genus are widespread in nature and found in soil, sewage, plants, and the intestinal tract of humans and animals. *K. pneumoniae* is a common pathogen, associated with a wide spectrum of infections, such as pneumonia, urinary tract infections, intra-abdominal infections, bloodstream infections, meningitis, and pyogenic liver abscesses [1]. The *Klebsiella* capsular polysaccharides are one of the major factors of virulence [2] and currently more than 80 variants of *Klebsiella* capsular types (K-types) are known [3]. Each K-type is associated with particular genes in the cluster of capsular synthesis (cps). The cps cluster has a mosaic structure with a group of six conserved genes (*galF, orf2, wzi, wza, wzb*, and *wzc*) at the 5’ end, which may be used for the molecular serotyping of *K. pneumoniae* strains [4,5]. Clinical isolates of *K. pneumoniae*, causing severe nosocomial infections, often possess multiple antibiotic resistances [6,7,8]. During the last decades, the prevalence of extended-spectrum β-lactamases (ESBL) producing *K. pneumoniae* has dramatically increased worldwide and *K. pneumoniae* is mostly associated with the dissemination of ESBLs and other horizontally transmissible resistance genes in most parts of the world [6,9]. A high fatality rate ranging between 18% and 49% has been reported for infections caused by multi-drug-resistant isolates of *K. pneumoniae* [1,10], and alternative therapeutics are required for to decrease this rate. One such alternative approach could be the use of lytic bacteriophages in the complex treatment of *Klebsiella* infections.

A large number of *Klebsiella* bacteriophages have been isolated and characterized and the GenBank database currently includes more than 120 genomes of *Klebsiella* phages. Of them, approximately 50 genomes belong to siphoviruses, one third are myoviruses, and a quarter are podoviruses. The majority of *Klebsiella* phages with a podovirus morphology belong to the *Przondovirus* and *Drulisvirus* genera within the Autographivirinae subfamily. To date, only one genome of N4-like *Klebsiella* phage Pylas is available in the Genbank database [MH899585]. Here, we describe a novel lytic N4-like bacteriophage KP8, specific to *K. pneumoniae*, including its genome, partial structural proteome, biological properties, and proposed taxonomy.

## 2. Materials and Methods

### 2.1. Bacterial Strains and Cultivation Media

The Enterobacteria strains used in this study were obtained from the Collection of Extremophilic Microorganisms and Type Cultures (CEMTC) of the Institute of Chemical Biology and Fundamental Medicine Siberian Branch of Russian Academy of Science and are listed in Appendix A. These strains were grown in Luria–Bertani (LB) medium and on plates containing LB-agar (1.5% *w*/*v*). All cultures were grown at 37 °C. Bacterial strains were identified by sequencing a 1308-bp fragment of the 16S rRNA gene and a 501 bp internal portion of the *rpoB* gene, as previously described [4]. To determine *K. pneumoniae* capsular types (K-types), PCR-amplification of a 580-bp fragment of the *wzi* gene was performed according to [5]. All PCR amplicons were gel purified (0.6% SeaKem^®^ GTG-agarose, Lonza, ME, USA) and sequenced (BigDye™Terminator v.3.1 Cycle Sequencing Kit and ABI 3500 Genetic Analyzer, Applied Biosystems, Foster City, CA, USA). The *wzi*-sequences of all investigated *K. pneumoniae* strains were deposited in the GenBank database under accession numbers MN371464–MN371513. The determined nucleotide sequences of 16S rRNA and *rpoB* gene fragments were compared with corresponding nucleotide sequences extracted from GenBank; *wzi* nucleotide sequences of the reference *Klebsiella* strains of the 77 K-types were extracted from the Supplementary file of Brisse et al. [5]. The online software MAFFT (https://mafft.cbrc.jp/alignment/server) was used for alignment of the sequences. Phylogenetic analysis was performed by the maximum-likelihood method with a bootstrap equal to 1000 in MEGA 6.0 software [11].

### 2.2. Phage Isolation and Propagation

Bacteriophage KP8 was isolated from a wastewater sample from the Novosibirsk infectious hospital №1. Bacteriophage isolation and propagation were performed as described previously [12], with slight modifications. To select bacteriophages, 10 mL of the sample was clarified by centrifugation at 10,000× *g* for 15 min and sterilized by filtration through a 0.22 μm filter (Millipore, Burlington, MA, USA). The filtrate was screened for bacteriophages by spotting 5–10 μL onto a fresh layer of *K. pneumoniae* CEMTC 356 in the top agar. The plates were incubated for 6 and 18 h at 37 °C, and each plaque was suspended in sterile phosphate buffer saline (PBS) to extract phage particles. Tenfold dilutions of obtained phage suspensions were spotted on a fresh layer of *K. pneumoniae* CEMTC 356 to obtain single phage plaques for subsequent phage extraction. The cycle of phage dilution and extraction was repeated three times.

Phage KP8 was propagated by infecting 100 mL of an exponentially growing culture of *K. pneumoniae* CEMTC 356 (optical density of 0.4 at 600 nm) at a multiplicity of infection (MOI, i.e., the ratio of phage to bacterium) of 0.01. Phage particles were purified from phage lysate by polyethylene glycol (PEG) precipitation, as described previously [13].

### 2.3. Phage Plaque Morphology and Electron Microscopy of Phage Particles

The morphology of plaques formed by phage KP8 on a layer of sensitive culture *K. pneumoniae* strain CEMTC 356 was determined using the double agar overlay method [14]. Plaques were examined after 18 h of incubation.

A drop of phage KP8 suspension with the titer of 109 plaque forming units per mL (PFU/mL) was adsorbed for 1 min on a copper grid covered with formvar film; then, the excess liquid was removed and the grid was contrasted on a drop of 1% uranyl acetate for 5–7 s. All of the samples were examined with a JEM 1400 transmission electron microscope (JEOL Ltd., Tokyo, Japan), and digital images were collected using a side-mounted Veleta digital camera (Olympus SIS, Münster, Germany).

### 2.4. Biological Properties of KP8 Bacteriophage and Host Range Analysis

All experiments on the biological properties of phage KP8 were performed twice in triplicate. A one-step growth curve and burst size experiments were carried out as described previously [15]. A lytic activity assay of phage KP8 was performed as previously described [16].

The infectivity of phage PM16 was defined for 74 Enterobacteria strains, including 63 *Klebsiella* strains (Appendix A). The host range was determined by the spotting of serial phage dilutions onto freshly prepared lawns of bacteria on agar plates, as described previously [17].

### 2.5. Phage DNA Purification and Sequencing

Phage DNA was extracted from the phage preparation as previously described [18]. To remove bacterial DNA and RNA, DNase and RNase (Thermo Fisher Scientific, Waltham, MA, USA) were added to the phage preparation to a final concentration of 10 µg/mL and the mixture was incubated for 2 h at 37 °C. Then, the phage suspension was supplemented with EDTA, proteinase K (Thermo Fisher Scientific, USA), and sodium dodecyl sulfate (SDS) to final concentrations of 20 mM, 100–200 µg/mL, and 0.5%, respectively, and the mixture was incubated for 3 h at 55 °C. After that, phage DNA was purified by phenol/chloroform extraction and subsequent ethanol precipitation.

KP8 genome library construction was performed using the Nextera DNA Sample Preparation Kit (Illumina Inc, San Diego, CA, USA). Sequencing was carried out using the MiSeq Benchtop Sequencer and MiSeq Reagent Kit v.1 (2 × 150 base reads). The genome was assembled de novo using CLC Genomics Workbench software v.6.0.1 and resulted in one genomic contig with an average coverage of 3364. The KP8 genome sequence was deposited in GenBank under accession number MG922974.

### 2.6. Genome Analysis

Putative open reading frames (ORFs) in the KP8 genome were identified using RAST [19]. In addition, the products encoded by the predicted ORFs were compared with the proteins deposited in the GenBank database, using the PSI-BLASTP algorithm. The predicted ORFs encoding hypothetical proteins identical to hypothetical phage ones and ORFs without homology with the sequences deposited in the GenBank database were analyzed by InterProScan and HHPred software [20,21]. The presence of tRNA genes was detected by tRNAscan-SE [22] and ARAGORN software [23]. Putative KP8 proteins were analyzed for the presence of trans-membrane helical domains and signal sequences using the Phobius [24] and TMHMM Server v. 2.0 software (http://www.cbs.dtu.dk/services/TMHMM).

The search for regulatory sequences was performed as follows. The promoter sequences were extracted from the *Escherichia* virus N4 genome [25] and compared with the KP8 genome sequence using Vector NTI software [26]. The most similar sequences found in the KP8 genome were proposed to be the promoter sequences. The FindTerm software version 2.8.1 (Softberry Inc.) was used for the search of potential phage ρ-independent terminator sequences in the KP8 genome.

A phylogenetic analysis of the KP8 phage genome and the most similar phage genomes was performed using IQ-Tree (http://iqtree.cibiv.univie.ac.at/) by the maximum-likelihood method. The most similar complete phage genomes were extracted from the GenBank database (http://www.ncbi.nih.gov) and sequence alignment was performed using Tcoffee software (http://tcoffee.crg.cat/). Mauve software [27] was used for a comparative analysis of the KP8 genome and genomes of the prototype *Escherichia* virus N4 [NC_008720], and *Escherichia* virus vB_EcoP_G7C [NC_015933]. The overall identity of aligned genomes was calculated using BioEdit software [28]. The average evolutionary divergence between the KP8 genome and phage genomes of *Enquatrovirus* and *Gamaleyavirus* genera was determined using MEGA v.6.0 [11].

A phylogenetic analysis of protein sequences was performed in a similar way using IQ-Tree. Protein sequences were obtained from the GenBank database. The nucleic acid substitution model LG + F + G4, and amino acid substitution model TVM + F + I + G4, as proposed by the IQ-Tree, were used. Edge support was assessed by the Bayes branch supports.

### 2.7. Phage Structural Proteins Analysis and Mass Spectrometry

Phage particles were purified from phage preparation by CsCl gradient ultracentrifugation [13]. Proteins from the purified phage KP8 particles were separated using Tris-glycine SDS 12% (*w*/*v*) polyacrylamide gel electrophoresis (PAGE) and visualized by Coomassie R250 and silver staining. Gel fragments containing individual protein bands were cut out from the gel and trypsin digestion was carried out, as described previously [29]. The obtained peptides were desalted using C18 ZipTips (Millipore), mixed (in a ratio of 0.5 μL of the sample to 0.5 μL of the matrix) with 2.5-dihydroxybenzoic acid dissolved in 70% acetonitrile and 0.1% trifluoroacetic acid, and spotted to a standard MTP ground steel plate (Bruker Daltonics, Billerica, MA, USA). Further analysis was performed using a MALDI-TOF spectrometer Ultraflex III (Bruker Daltonics). The mass spectra of protein tryptic digests were recorded in a reflective positive ion mode in the 500–4200 *m*/*z* range. The mass spectra were processed and analyzed using the mMass v.5.5.0 software [30]. Database searches were performed against the NCBIprot database using Mascot Peptide Mass Fingerprint (http://www.matrixscience.com).

## 3. Results and Discussion

### 3.1. Isolation, Morphology, and Host Range

The strain *K. pneumoniae* CEMTC 356 was used as a host strain for isolation of the bacteriophage KP8. The K-type for this strain was determined using wzi gene sequencing, which is a rapid method suggested previously [5]. Phylogenetic analysis showed that the CEMTC 356 *wzi* gene sequence differed from all available *wzi* genes of *Klebsiella* reference strains (Figure 1), so the host strain had a K-type different from previously found K-types.

The KP8 phage formed clear plaques, with a diameter of about 1–1.5 mm on a lawn of the host strain (Figure 2A). Electron microscopy revealed phage capsids with a diameter of 50–55 nm connected with a short tail of approximately 16 nm in length (Figure 2B). The morphology and size of the phage particles corresponded to those of podoviruses [31]. The host range of phage KP8 was determined using 74 Enterobacteria strains (Appendix A), including 63 *Klebsiella* strains, but it was only specific to *K. pneumoniae* CEMTC 356.

### 3.2. Biological Properties of KP8

A one-step growth experiment revealed a latent period of 15 min with a burst size of ~40 phage particles per infected cell (Figure 3A). The multistep bacterial killing curve for phage KP8 showed that the amount of living bacteria decreased dramatically in 2 h after infection, and afterwards slowly increased (Figure 3B). The results indicated that phage KP8 was a lytic phage.

### 3.3. Genome Structure

The size of the KP8 genome was 73,679 bp, with a 404 bp length of the terminal direct repeats (Figure 4). The GC-content in the KP8 genome was 44.2%, in contrast to its host, *Klebsiella pneumoniae*, which has a GC-content of about 57%. The KP8 genome demonstrated the highest similarity to the genomes of *Enquatrovirus* and *Gamaleyavirus* phages, which are N4-like podoviruses. The nucleotide sequence identity of the KP8 genome to those of phages representing these two genera was 46.6% for *Escherichia* phage N4 (*Enquatrovirus*), and 46.8% for phage G7C (*Gamaleyavirus*). In addition, the KP8 genome showed gene synteny typical for the members of these two genera (Figure 5) and contained the gene, encoding a large virion-encapsulated RNA polymerase (vRNAP).

The genome of KP8 comprised 97 putative ORFs and most of them were organized into two large functional clusters. The first cluster (ORF1–ORF67) contained early and middle genes, while the second cluster (ORF68–ORF87) included late genes that were transcribed in the opposite direction (Figure 4). In addition, ten hypothetical ORFs were located at the 3’ end of the KP8 genome, which was transcribed in the right direction. ATG was proposed as a start-codon for 94 ORFs and only three ORFs used GTG as a start-codon. Four potential promoters (P1–P4) were found at the 5’ end of the KP8 genome. Notably, P3 and P4 sequences were revealed before the genes, encoding RNA-polymerase 1 subunits A and B, respectively (Table 1). A potential rho-independent terminator was identified at the end of the first cluster (Table 1).

### 3.4. Predicted Gene Products

From 97 putative ORFs of KP8, 40 ORFs encoded products with predicted functions, and 57 ORFs encoded hypothetical proteins. ORFs providing RNA and DNA synthesis were identified in the first cluster of genes. The RNAP1 subunit A, a predicted analog of gp2 in the N4 phage, was encoded by the ORF5 located right after the putative promoter P3, while the RNAP1 subunit B was encoded by the ORF22 that was located after the putative promoter P4. In a group of putative ORFs located between ORF5 and ORF22, two ORFs were responsible for the predicted functions—the GTG-binding domain (gp14) and ADP-ribosylglycohydrolase (gp19), respectively. ORF25 (RNAP2) was followed by ORF26, encoding a potential capsid decorating protein with an Ig-like domain (gp26), which was found in the first cluster that is usual for N4-like phages. The sub-cluster of ORFs associated with DNA synthesis was typically located downstream the sub-cluster of ORFs providing RNA synthesis (Figure 4, Table 2). In addition, six tRNA genes were grouped at the 3’ end of the first cluster and four of them were identified as the asparagine (GTT), proline (CCA), leucine (CAG), and glutamine (TTG) tRNA genes (Figure 4).

Within the second cluster, ORFs encoding vRNAP and structural proteins and ORFs associated with the assembly of phage capsids, DNA maturation, and the outburst of mature phage particles were predicted. A group of predicted structural KP8 proteins was similar to that of N4 and G7C phages (Figure 5, Table 2), except gp83 and gp84. Both products were annotated as predicted tail fiber proteins with hydrolyzing activity domains (Table 2). Notably, none of the proteins similar to gp83 were found in known N4-like phages. The gp83 of KP8 showed the highest similarity (23% of identity per 78% coverage) to the tailspike protein [YP_009153195] of *Klebsiella* myovirus KP64-1 [NC_027399.1]. Previously, depolymerizing activity of this tail spike protein has been experimentally confirmed [32]. In the gp84 of KP8, the SGNH hydrolase domain (aa 340–579) was predicted using InterProScan. The N-terminal domain was similar to that of tail fiber proteins of several N4-like phages and, presumably, this region is responsible for the attachment of the tail fiber to the phage capsid.

Transmembrane helical domains and signal sequences were found in 13 of the predicted proteins (Appendix A), and the majority of appropriate ORFs encoded hypothetical proteins, except ORF78, ORF79, and ORF80, which were associated with the release of phages from the cell. According to our analysis, the N-acetylmuramidase (ORF79) contained a signal sequence and this protein was proposed to be SAR-endolysin. Notably, four transmembrane domains were predicted in a hypothetical protein encoded by ORF71. As this ORF was located in the cluster of genes encoding structural proteins, we suppose that it might encode a structural protein.

### 3.5. Proteomic Analysis

In addition, a proteomic analysis of virion proteins was carried out. At least nine protein bands were identified using SDS-PAGE (Figure 6). Seven protein bands were subjected to MALDI-TOF analysis. The obtained results confirmed the genomic annotation of vRNAP and six structural proteins with the predicted major coat protein, portal protein, capsid decoration protein, and hypothetical structural protein, and unusual gp83 and gp84 (Table 2). Some putative structural proteins were not identified by this method, which was probably due to their relatively low abundance.

### 3.6. Phylogenetic Analysis of the KP8 Complete Genome

To estimate the taxonomic position of *Klebsiella* bacteriophage KP8, a phylogenetic analysis of the complete KP8 genome, along with the most similar phage genomes, was carried out. It was revealed that on the phylogenetic tree of the N4-like group of phages, the KP8 genome formed a distinct branch within the clade, which included the members of *Enquatrovirus* and *Gamaleyavirus* genera, besides KP8 (Figure 7). Importantly, the KP8 genome did not group with the known *Klebsiella* phage Pylas, which probably belongs to the *Ithacavirus* genus (Figure 7).

As the most similar genomes of N4-like phages belonged to the members of *Enquatrovirus* and *Gamaleyavirus* genera, the average evolutionary divergence over sequence pairs was estimated between the KP8 genome and phage genomes of *Enquatrovirus*, and *Gamaleyavirus* genera, respectively. As a result, the evolutionary divergence of KP8/*Enquatrovirus* was 0.466 substitutions per site (substitutions/site) and that of KP8/*Gamaleyavirus* was 0.447 substitutions/site. This was quite similar to the evolutionary divergence between *Enquatrovirus* and *Gamaleyavirus* genera (0.468 substitutions/site). Note that the evolutionary divergence within *Enquatrovirus* and *Gamaleyavirus* genera was 0.017 and 0.117 substitutions/site, respectively. As the evolutionary divergence between KP8 and the *Klebsiella* phage Pylas was high, with a value of 0.963 substitutions/site, the KP8 phage might be the first member of a new genus within N4-like phages and this genus did not include *Klebsiella* phage Pylas.

To confirm the result of the phylogenetic analysis of complete genomes, amino acid sequences of some predicted proteins encoded by the KP8 genome (primase, ORF55; vRNAP, ORF68; major coat protein, ORF74; terminase, ORF86) were compared with the most similar phage protein sequences, downloaded from the Genbank database (Appendix A). All visualized phylogenetic protein trees confirmed the position of the KP8 proteins on a distinct branch in the clade containing appropriate proteins of phages from the *Enquatrovirus* and *Gamaleyavirus* genera. The obtained data suggested that *Klebsiella* phage KP8 differs from other similar phages and may represent a new genus within the N4-like phages.

## 4. Discussion

N4-like podoviruses comprise a unique group among bacteriophages because their virions include the giant vRNAP that enters the host-cell during phage infection and, therefore, N4-like phages do not require the host RNA-polymerase for transcription of their early genes [25,33]. Over the last decade, tens of N4-like phages have been discovered and their genomes have been sequenced and characterized [33,34,35,36,37,38,39,40,41,42,43,44,45,46]. They have conserved genome organization, including two large clusters of early-middle and late genes, transcribed in opposite orientations, and a group of usually short ORFs with unknown functions, whose number varies between different N4-like phages. Currently, N4-like phages capable of infecting various Alpha-, Beta-, and Gamma-proteobacteria have been described [37,40,42,43,44,45,46] and many hosts from the Gamma-proteobacteria class are human pathogens, including the ESKAPE group. This fact explains the interest in N4-like phages, all of which are strictly lytic phages.

*K. pneumoniae*, which is currently recognized as super-bacteria, is a host of two different *Klebsiella* N4-like phages: *Klebsiella* phage Pylas, isolated in the USA, Texas [MH899585]; and the novel KP8 phage described in this article. The KP8 phage is a lytic phage infecting *K. pneumoniae* CEMTC 356 that does not belong to the most virulent K1 and K2 capsular types. The KP8 genome shows gene synteny typical for the group of N4-like phages and contains the gene encoding vRNAP. According to phylogenetic analysis of the complete KP8 genome, along with the most similar phage genomes, it was revealed that KP8 forms a distinct branch within the N4-like group of phages. Notably, the evolutionary divergence between KP8 and *Klebsiella* phage Pylas was higher than that between KP8 and other genera, including *Enquatrovirus* and *Gamaleyavirus*. This fact confirmed that KP8 and *Klebsiella* phage Pylas definitely belongs to different genera and *K. pneumoniae*, along with *Escherichia coli* and *Erwinia amylovora*, can be infected with N4-like phages from different genera. Finally, taking into consideration (i) the unique feature of all N4-like phages—the presence of vRNAP in a capsid that distinguishes this group of phages from the phage world; (ii) the comparatively high genetic divergence of N4-like pages, which currently form more than ten genera; and (iii) the wide range of hosts for these phages, we strongly support the assignment of N4-like phages to a separate subfamily, presumably called “Enquartavirinae”, as was suggested previously [47].

## Figures and Tables

**Figure 1 viruses-11-01115-f001:**
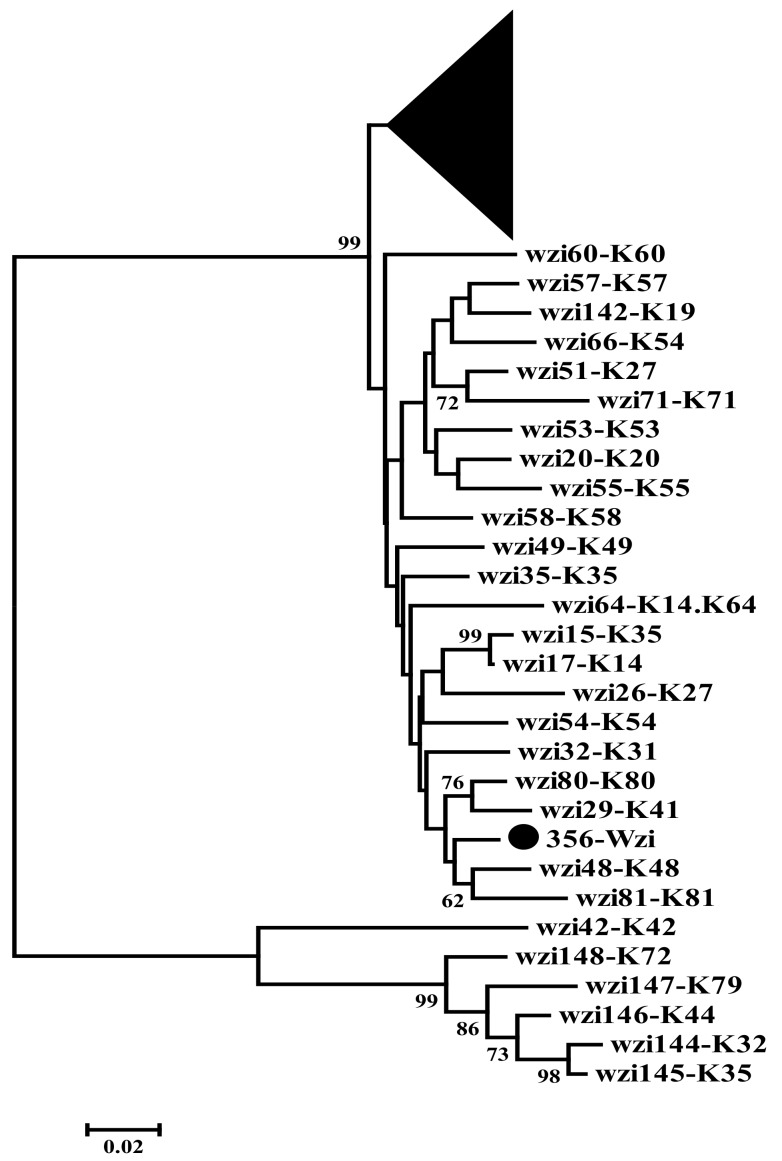
Phylogenetic analysis of wzi nucleotide sequences of the *Klebsiella* strain CEMTC 356 and reference *Klebsiella* strains of the 77 K-types. Each serotype is denoted by the letter K and the number following it [5]. The *wzi* sequence of the investigated CEMTC 356 is marked with a black circle. The compressed subtree includes more distant reference sequences. The maximum-likelihood method was used to construct the tree with a bootstrap equal to 1000. Statistical support above 70% is shown at the nodes.

**Figure 2 viruses-11-01115-f002:**
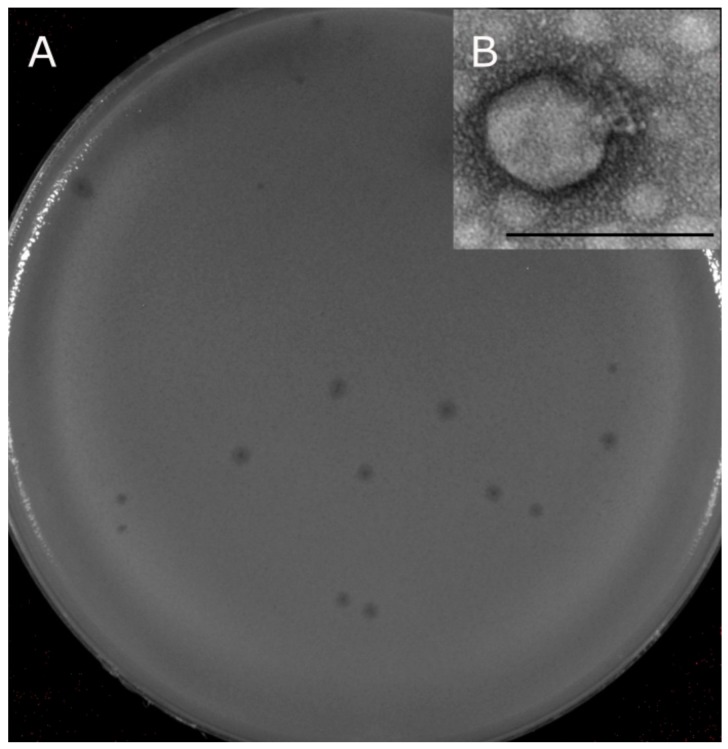
(**A**) Plaques formed by phage KP8 on a lawn of the host strain CEMTC 356; (**B**) electron microphotograph of phage particle. The scale corresponds to 100 nm.

**Figure 3 viruses-11-01115-f003:**
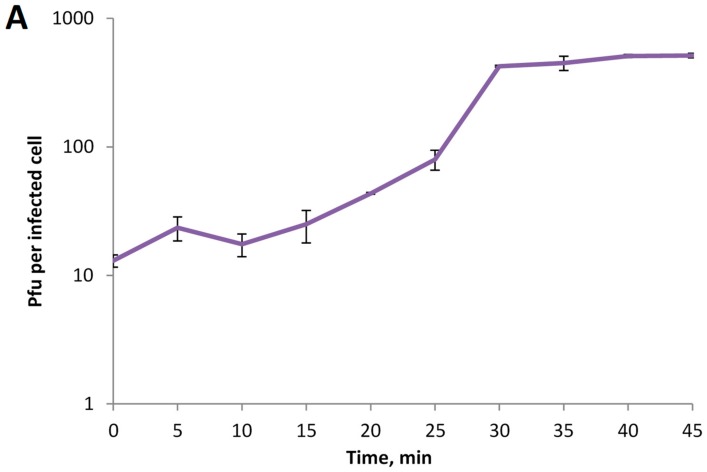
Biological properties of phage KP8. (**A**) One-step growth curve of phage KP8; (**B**) multistep bacterial killing curve of phage KPM16; intact growing *K. pneumoniae* CEMTC 356 cells were used as a control.

**Figure 4 viruses-11-01115-f004:**
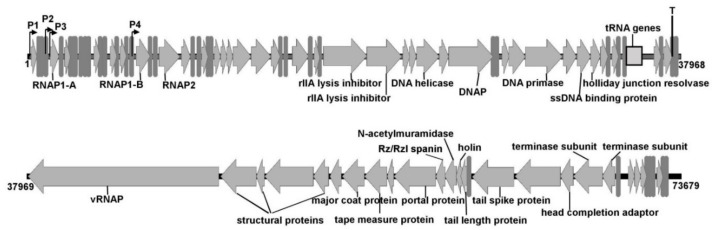
A schematic genomic map of bacteriophage KP8. A number of genes with proposed functions are noted.

**Figure 5 viruses-11-01115-f005:**
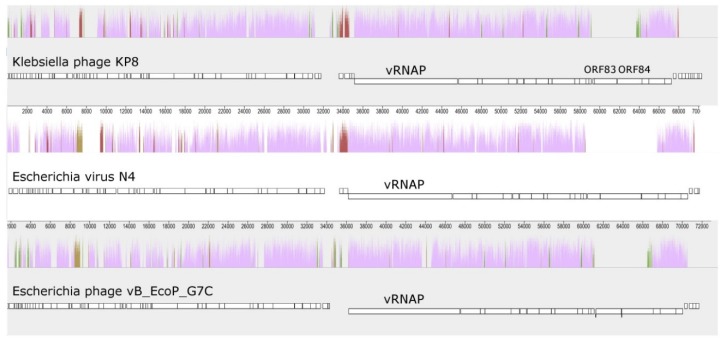
Gene synteny of KP8, *Escherichia* virus N4, and *Escherichia* phage vB_EcoP_G7C genomes. The alignment was performed using Mauve software with default parameters. Horizontal white bars match the genomic sequences with rectangles corresponding to the ORFs. The height of the colored peaks in the chromatograms represents the level of similarity in the particular region between aligned genomes. The light purple regions in the chromatograms are similar in all genomes. Brown, green, and marsh-colored regions correspond to KP8/N4, KP8/G7C, and N4/G7C similarity of genomes, respectively.

**Figure 6 viruses-11-01115-f006:**
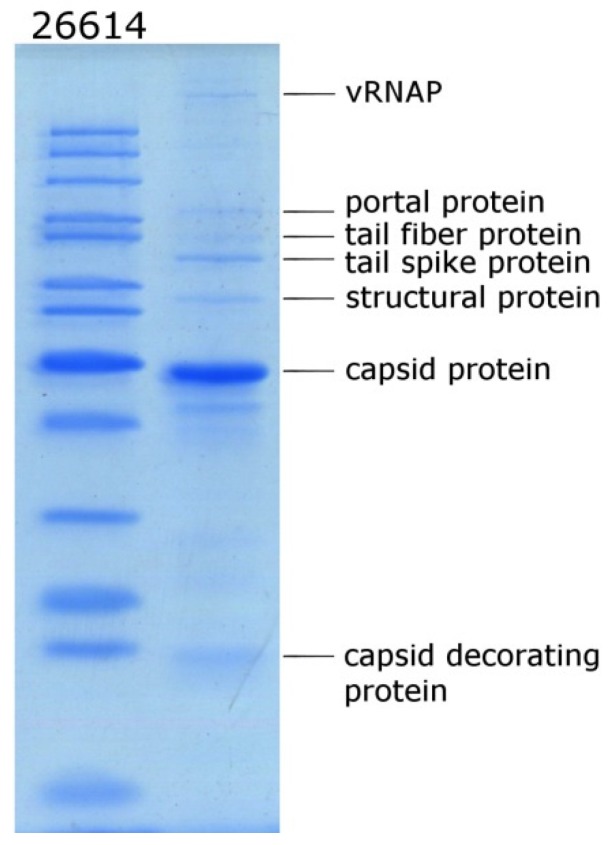
SDS-PAGE of purified bacteriophage KP8 particles, followed by staining with Coomassie brilliant blue R250. Lane 26614 is unstained protein standards (Thermo Fisher Scientific, USA).

**Figure 7 viruses-11-01115-f007:**
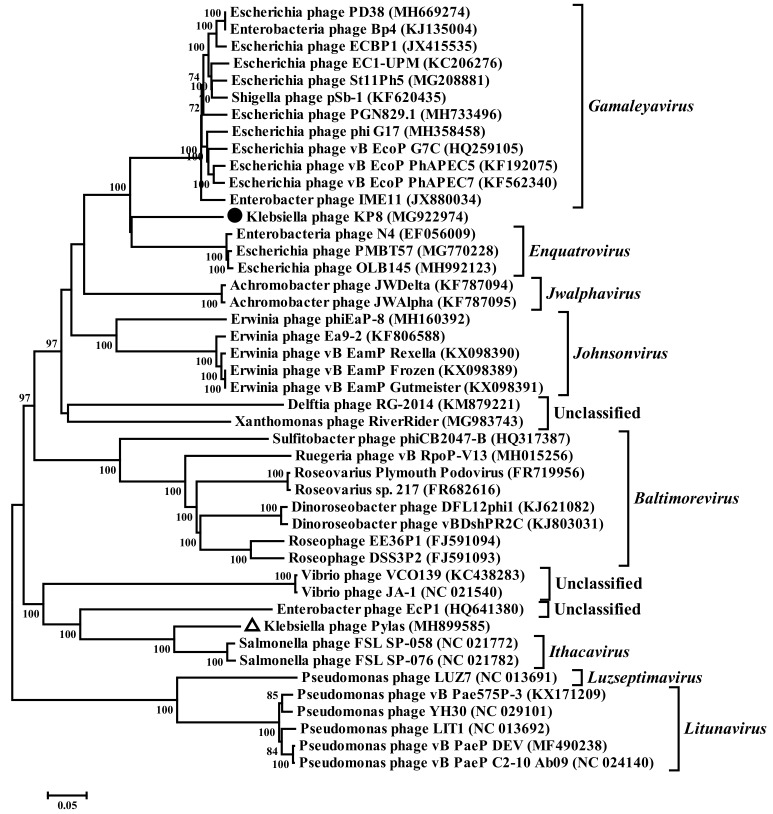
Phylogenetic analysis of the complete genome of KP8 bacteriophage along with the most similar genomes. GenBank identifiers (gi) for the sequences are shown in parentheses. The maximum-likelihood method was used to construct the tree. Statistical support above 70% is shown at the nodes. The sequence ID of the genome of the investigated KP8 phage is marked with a black circle. The sequence ID of the *Klebsiella* phage Pylas is marked with a triangle.

**Table 1 viruses-11-01115-t001:** Putative regulatory elements in the genome of *Klebsiella* phage KP8.

Name	Position	ORF ^a^	Regulatory Element Sequence ^b^
Putative promoter			
P1	89–110	1	AC***GTTG***CTCCG***CAAC***CTATGGA
P2	962–983	4	AA***GACG***CTCCG***CGTC***TTATGGA
P3	1225–1246	5	G***GGCTG***CTCCG***CAGCC***TTTGGA
P4	5965–5984	21	AA***GCAG***CTCCG***CTGC***TTGTGGG
Putative ρ-independent terminator			
T	37311–37349	64	TGCTGAAGA**ATGGGCAAGTAG**TGCCTT**AGAGCGACCTTG**

^a^ The number of downstream ORFs is given for promoters and the number of upstream ORF for terminator. ^b^ Bold italics denote palindromic sequences in putative promoter sequences. Start of transcription is denoted by bold. Underlined bold sequences are palindromic sequences for terminator.

**Table 2 viruses-11-01115-t002:** Genes with proposed functions in the genome of bacteriophage KP8.

No	ORF	ORF Position (b.p.)	Length of Product (aa)/Predicted Molecular Mass (kDa)	Start Codon/ORF Orientation (+−)	Maximal Identity (%) with GenBank Phage Protein Sequences, according to Algorithm BLASTXa	Predicted Molecular Function	Protein Sequence Coverage, (%) in MS-Analysis
1	5	1254–1748	165/18.2	ATG/+	74.2 (*Escherichia* virus N4 [YP_950480])	RNA polymerase 1 subunit A	
2	14	3799–4155	119/13.3	ATG/+	75.4 (*Escherichia* phage EC1-UPM [YP_009598285])	GTP binding protein DUF2493/DNA processing A domain	
3	19	5304–5684	126/13.89	ATG/+	77.0 (*Escherichia* virus N4 [YP_950492])	ADP-ribosylglycohydrolase	
4	22	6257–7072	272/31.73	ATG/+	74.6 (Escherichia virus N4 [YP_950493])	RNA polymerase 1 subunit B	
5	25	7589–8803	404/45.99	ATG/+	74.1 (*Escherichia* phage PMBT57 [AUV59092])	RNA polymerase 2 subunit A	
6	26	8896–9411	171/17.95	ATG/+	48.5 (*Erwinia* phage phiEaP-8 [AWN06248])	Capsid decorating protein (Ig-like domain)	84
7	29	10082–10432	116/13.24	ATG/+	88.4 (*Escherichia* virus N4 [YP_950500])	HNH-endonuclease	
8	35	11890–12942	350/35.9	ATG/+	63.0 (*Escherichia* phage OLB145 [AYR04207])	ATPase	
9	36	12953–14104	384/43.8	ATG/+	62.9 (*Escherichia* phage OLB145 [AYR04209])	Metallopeptidase	
10	37	14112–14639	175/19.5	ATG/+	78.7 (*Escherichia* phage phiG17 [AWY03411])	dCTP deaminase	
11	40	15353–16294	313/35.9	ATG/+	67.7 (*Escherichia* phage PMBT57 [AUV59106])	FAD-dependent thymidylate synthase	
12	44	17167–19689	840/95.1	ATG/+	43.5 (*Escherichia* virus N4 [YP_950511])	rIIA lysis inhibitor	
13	45	19686–21722	678/73.69	ATG/+	52.7 (*Escherichia* virus N4 [YP_950512])	rIIB lysis inhibitor	
14	46	21782–22177	131/14.9	ATG/+	84.6 (*Escherichia* phage PMBT57 [AUV59111])	DNA-binding domain^1^	
15	47	22158–22562	134/14.8	GTG/+	64.4 (*Escherichia* phage vB_EcoP_PhAPEC5 [YP_009055545])	NTP pyrophosphohydrolase	
16	48	22599–23906	435/49.05	ATG/+	71.3 (*Escherichia* virus N4 [YP_950515])	DNA-helicase	
17	50	24454–27036	860/97.5	ATG/+	74.8 (*Escherichia* phage vB_EcoP_G7C [YP_004782168.1])	DNA-polymerase I	
18	51	27036–27251	71/7.88	ATG/+	47.0 (*Klebsiella* phage Soft [QEM42165.1])	Nucleoside/nucleotide kinase^1^	
19	53	27556–27966	136/15.34	ATG/+	79.8 (*Escherichia* virus N4 [YP_950519.1])	3’-phosphatase 5’-polynucleotide kinase	
20	54	27950–28921	323/37.33	GTG/+	78.0 (*Escherichia* virus N4 [YP_950520.1])	PD-(D/E)XK nuclease superfamily protein	
21	55	28921–31068	715/81.8	ATG/+	80.7 (*Escherichia* phage PMBT57 [AUV59047])	DNA primase	
22	56	31126–31884	252/28.68	ATG/+	86.9 (*Escherichia* phage PGN829.1 [AXY82607.1])	Nucleoside triphosphate hydrolase	
23	57	31927–32721	264/28.45	ATG/+	57.7 (*Escherichia* virus N4 [YP_950523.1])	ssDNA-binding protein	
24	58	32721–33278	185/20.3	ATG/+	73.5 (*Escherichia* virus N4 [YP_950524.1])	Holliday junction resolvase	
25	68	37968–48392	3474/379.07	ATG/−	69.9 (*Escherichia* phage OLB145 [AYR04234.1])	Virion RNA polymerase	8
26	69	48494–50443	649/70	ATG/−	69.3 (*Escherichia* phage vB_EcoP_PhAPEC7 [YP_009056187.1])	Putative structural protein	78
27	70	50453–50893	146/15.5	ATG/−	66.4 (*Escherichia* phage ECBP1 [YP_006908829])	Putative structural protein	
28	72	53569–54399	276/30.3	ATG/−	85.6 (*Escherichia* phage IME11 [YP_006990617])	Putative structural protein	
29	74	55147–56352	401/44.1	ATG/−	90.5 (*Escherichia* phage vB_EcoP_PhAPEC5 [YP_009055570.1)	Major coat protein	91
30	75	56365–57594	409/45	ATG/−	66.3 (*Escherichia* phage IME11 [YP_006990614])	Tape measure protein	
31	77	57976–60261	761/85.1	ATG/−	76.9 (*Escherichia* phage PGN829 [AXY82589])	Portal protein	57
32	78	60270–60767	165/18.8	ATG/−	65.9 (*Escherichia* phage IME11 [YP_006990611])	Rz/RzI spanin protein	
33	79	60751–61389	212/22.8	ATG/−	81.8 (*Escherichia* phage vB_EcoP_PhAPEC7 [YP_009056197])	N-acetylmuramidase	
34	80	61373–61699	108/12.6	ATG/−	56.6 (*Shigella* phage pSb-1 [YP_009008416])	Holin	
35	81	61624–61956	110/12.3	ATG/−	97.8 (*Escherichia* phage IME11 [YP_006990608])	Tail length tape-measure protein	
36	83	62259–64526	755/82.25	ATG/−	23.7 (*Klebsiella* phage K64-1 [YP_009153195])	Tail spike protein/endo-N-acetylneuraminidase	64
37	84	64598–67075	825/89.3	ATG/−	30.1 (*Escherichia* phage IME11 [YP_006990695])	Tail fiber protein/SGNH-esterase domain ^1^	56
38	85	67072–67779	235/61	ATG/−	97.0 (*Escherichia* phage IME11 [YP_006990694])	Head completion adaptor	
39	86	67786–69378	530/60.7	ATG/−	89.4 (*Escherichia* virus N4 [YP_950546])	Terminase large subunit	
40	87	69371–70063	230/25.6	ATG/−	86.1 (*Escherichia* virus N4 [YP_950547])	Terminase small subunit	

^1^ ORFs, which predicted molecular function, were identified using InterProScan or HHPred.

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
