# Peer review of "Isolation and Characterization of a Novel Klebsiella pneumoniae N4-like Bacteriophage KP8"

_viruses, 2019, doi:10.3390/v11121115_

Round 1
Reviewer 1 Report
This paper deals with the discovery of a new phage, infecting K.pneumoniae related to the 77 K-types. The manuscript is well written, accurate and sound. The results are relevant both for the practical aspects and for the contribution to the knowledge about N4-like bacteriophages. Due to the recent outbreaks of Klebsiella clonally related strains, a further MLST characterization, at least of the host strain, would bring additional value to this nice work. see https://pubmlst.org/kpneumoniae/info/primers.shtml
Minor points: some of the bacterial names are not in italics
Author Response
Dear Reviewer,
We would like to thank the Reviewer for the positive response and comments that are very important and useful for our future research. Indeed, MLST is a powerful tool for analysis of distribution clonally related microbial strains and it is used mostly in epidemiology research. In our research, we hoped to predict possible receptor for the KP8 phage, as they believe that the majority of receptors for Klebsiella podoviruses are capsule polysaccharides. However, after K. pneumoniae capsular type (K-type) determination, we found that new K-type, different from previously found. So, additional investigation is required to determine the receptor. Unfortunately, MLST analysis does not include cps cluster (one serotype may include different clonal types; please, see Liao et al. Capsular serotypes and multilocus sequence types of bacteremic Klebsiella pneumoniae isolates associated with different types of infections. Eur J Clin Microbiol Infect Dis. 2014 Mar;33(3):365-9. doi: 10.1007/s10096-013-1964-z) and could not be helpful in this case. We plan to use MLST for the characterization of Klebsiella strains from our collection.
Minor points: some of the bacterial names are not in italics:
Bacterial names were corrected.
Reviewer 2 Report
The authors describe the isolation and characterization of a new lytic virus belonging to the Podoviridae family. The authors present their study and data thoroughly and efficiently. I find no changes needed for publications.
Not contingent on publication, however I did find ORF83/84 interesting and wondered if these could possibly be depolymerases (see perhaps Lin, 2014). As they are often annotated as tail fiber proteins, as is shown in this manuscript, could the authors provide information about the possibility of these proteins perhaps being depolymerases. Also, the distinctive halo around plagues does seem evident in Figure 2, although somewhat unclear.
Author Response
Dear Reviewer,
We are grateful for a positive response to our manuscript and interesting comment, concerning ORF83/84 putative depolymerizing activity.
The putative functions of ORF83/ORF84 were predicted according to blastx search. The most similar annotated proteins to gp83 were tail spike protein of phage K64-1 (YP_009153195.1) and a number of endo-N-acetylneuraminidases, encoded by genomes of Klebsiella strains (BAS39799.1; SLR96181.1; STU22633.1). Previously, the tail-spike protein of phage K64-1 has been experimentally shown to have depolymerizing acitivity (please, see Yi-Jiun Pan et al. Identification of Capsular Types in Carbapenem-Resistant Klebsiella pneumoniae Strains by wzc Sequencing and Implications for Capsule Depolymerase Treatment. Antimicrob Agents Chemother. 2015 Feb; 59(2): 1038–1047. doi: 10.1128/AAC.03560-14). As for gp84, the most similar annotated protein was endo-alpha-sialidase of Escherichia coli (GDC59608.1). In addition, the depolymerizing activity of the protein encoded by ORF84 was confirmed using InterProScan, and the SGNH hydrolase domain was predicted in 340-579 amino acids. We added this information to the manuscript (please, see page 10, Lanes 256-260, marked with bold).